# Prescription Infant Formulas Are Contaminated with Aluminium

**DOI:** 10.3390/ijerph16050899

**Published:** 2019-03-12

**Authors:** James Redgrove, Isabel Rodriguez, Subramanian Mahadevan-Bava, Christopher Exley

**Affiliations:** 1Life Sciences, Huxley Building, Keele University, Staffordshire ST5 5BG, UK; jamesredgrove11@gmail.com; 2The Birchall Centre, Lennard-Jones Laboratories, Keele University, Staffordshire ST5 5BG, UK; i.rodriguez.nunez-milara@keele.ac.uk; 3Russells Hall Hospital, Dudley Group Foundation NHS Trust, Pensnett Road, Dudley DY1 2HQ, West Midlands, UK; s.mahadevan@nhs.net

**Keywords:** aluminium contamination, infant formulas, infant nutirion, aluminium toxicity, human exposure to aluminium

## Abstract

Historical and recent data demonstrate that off-the-shelf infant formulas are heavily contaminated with aluminium. The origin of this contamination remains to be elucidated though may be imported via ingredients, packaging and processing. Specialised infant formulas exist to address health issues, such as low birth weight, allergy or intolerance and medical conditions, such as renal insufficiency. The aluminium content of these prescription infant formulas is measured here for the first time. We obtained 24 prescription infant formulas through a paediatric clinic and measured their total aluminium content by transversely heated graphite furnace atomic absorption spectrometry following microwave assisted acid/peroxide digestion. The aluminium content of ready-to-drink formulas ranged from 49.9 (33.7) to 1956.3 (111.0) μg/L. The most heavily contaminated products were those designed as nutritional supplements for infants struggling to gain weight. The aluminium content of powdered formulas ranged from 0.27 (0.04) to 3.27 (0.19) μg/g. The most heavily contaminated products tended to be those addressing allergies and intolerance. Prescription infant formulas are contaminated with aluminium. Ready-made formulas available as nutritional supplements to aid infant growth contained some of the highest concentrations of aluminium in infant formulas measured in our laboratory. However, a number of prescription infant formulas contained the lowest concentrations of aluminium yet measured in our laboratory. These higher cost specialist preparations demonstrate that the contamination of infant formulas by aluminium is not inevitable. They represent what is achievable should manufacturers wish to address the threat posed to health through infant exposure to aluminium.

## 1. Introduction

It is five years since we last reported the significant contamination of infant formulas by aluminium [1,2]. Recent research, though limited in its scope, suggests that off-the-shelf formulas remain heavily contaminated [3]. There exists a wide range of specialised infant formulas that are often only available through paediatric clinics and prescription. These are designed to address a number of nutritional issues including low birth weight, perceived intolerances, gastrointestinal disorders, allergies and renal insufficiency [4]. Many of these products are fed to vulnerable infants under the expected guidance of a paediatrician. Some may be combined with medication [5].

Human exposure to aluminium is a serious health concern [6]. Aluminium exposure in infants is understandably a burgeoning issue [7,8]. While infant exposure to aluminium continues to be documented, its consequences, immediate and in the future, have received only scant attention [1,2] and research is required to understand the biological availability of aluminium through formula feeding. For example, how much aluminium is absorbed across the neonate gut and its subsequent fate, including excretion.

There is already too much aluminium in infant formulas [1,2] and herein we have measured its content in a large number of prescription formulas, products which are fed to vulnerable infants in their first months of life. Many of these products are heavily contaminated with aluminium.

## 2. Materials and Methods

We obtained 24 prescription infant formulas through the Paediatric Clinic of Russells Hall Hospital, Dudley, United Kingdom. Both ready-to-drink and powdered products were supplied as pristine, unopened samples. They included ready-made drinks for preterm infants and those having intrauterine growth restriction (IUGR), supplements in the form of ready-made drinks for infants having poor weight gain, powdered formulas for allergy and intolerance and powdered formulas with additional amino acids (see Table 1, Table 2, Table 3, Table 4 and Table 5 for brand names). 

Each unopened product (to avoid potential extraneous contamination) was mixed manually before being opened and sampled according to needs. The total aluminium content of all formulas was measured by transversely heated graphite furnace atomic absorption spectrometry (TH GFAAS) following acid/peroxide microwave digestion. Analytical methods and quality assurance data are identical to those used previously in our laboratory [1,2,9] and so are not detailed here. Data are presented according to product specialisation (Table 1, Table 2, Table 3 and Table 4) and by way of comparing ready-made and powdered formulations (Table 5).

## 3. Results

### 3.1. Ready-Made Drinks for Preterm and IUGR Infants

The concentration of aluminium (mean and SD) ranged from 49.9 (33.7) to 249.4 (64.0) μg/L while the amount of aluminium per serving varied from 3.5 to 45.7 μg depending upon serving volume (Table 1). The %RSD (relative standard deviation) was consistently high across all products and probably reflects the inhomogeneous nature of the milks and the non-uniform distribution of aluminium throughout the bulk volume.

### 3.2. Ready-Made Drinks as Supplements for Weight Gain

The concentration of aluminium (mean and SD) ranged from 153.5 (161.3) to 1956.3 (111.0) μg/L while the amount of aluminium per serving varied from 25.6 to 391.3 μg depending upon serving volume (Table 2). Again the %RSD (relative standard deviation) was high across all but one product and probably demonstrates the uneven distribution of aluminium throughout the bulk volume of a product.

### 3.3. Powdered Formulas for Allergies and Intolerance

The concentration of aluminium (mean and SD) in the powders ranged from 0.35 (0.03) to 3.27 (0.19) μg/g (Table 3). The amount of aluminium per serving varied from approximately 4–71 μg at birth to 12–92 μg at six months of age. Where data were available aluminium per day ranged from 26–231 μg at birth to 47–367 μg at six months of age. The %RSD (relative standard deviation) for these products were not especially high which suggested a more even distribution of contaminating aluminium in powdered products.

### 3.4. Powdered Formulas with Additional Amino Acids

The concentration of aluminium (mean and SD) in the powders ranged from 0.27 (0.04) to 2.23 (1.23) μg/g (Table 4). The amount of aluminium per serving varied from approximately 4–28 μg at birth to 8–64 μg at six months of age. Where data were available aluminium per day ranged from 21–167 μg at birth to 24–256 μg at six months of age. The %RSD (relative standard deviation) for these products were not especially high which suggested a more even distribution of contaminating aluminium in powdered products.

## 4. Discussion

Prescription infant formulas are contaminated with aluminium. Among the ready-made milks those prescribed as supplements to aid slow growth rate (Table 2) were, with few exceptions, significantly more contaminated than those for pre-term or IUGR infants (Table 1). The Nutricia Fortini range of products was consistently high in aluminium with concentrations between 500 and 800 μg/L. One apple-flavoured product from Abbott Nutrition was contaminated to a level of 2 mg/L aluminium. For the powdered formulas, those with additional amino acids (Table 4) contained less aluminium than those designed for allergies and intolerance (Table 3). The Nutramigen Puramino product was an exception to this rule, while another Nutramigen product (Pregestimil Lipil) was also the most contaminated of the allergy formulas. When the aluminium contents of all products as ready-to-use formulas are compared it is interesting to note that powdered products are generally less contaminated than ready-to-drink products (Table 5). This distinguishes this group of prescription formulas from previous off-the-shelf products where the powdered forms were found to contain the highest contents of aluminium [2,3]. Intriguingly some of the prescription formulas measured herein were lower in aluminium content (e.g., 41.4 (6.1) to 67.5 (20.5) μg/L) than any other formula product measured previously in our laboratory (Table 5). This may be indicative that the contamination of infant formulas by aluminium is not inevitable. It may suggest that selected ingredients added to premium products can reduce contamination by aluminium and, apparently, irrespective of the aluminium-based packaging used in all these products. Since all manufacturers of infant formulas deny the knowing addition of aluminium to their products, it remains a mystery as to its source. The ingredients supplied to infant formula manufacturers are likely sources of aluminium contamination. For example, we recently measured the aluminium content of whey protein hydrolysates (on behalf of a major manufacturer of such products) and found they contained between 4.1 and 8.1 μg/g aluminium. This represents one ingredient of infant formulas that could be contributing significant amounts of aluminium to the final product. In the products measured herein and especially the ready-to-drink supplements (Table 2) it is clear that the inclusion of fruit or fruit flavourings may be importing aluminium into the final product. Finally, the equipment used in processing of formulas could be a significant source of contamination and especially if the containers and utensils used in these operations are aluminium-based.

## 5. Conclusions

Aluminium is toxic in humans [10]. There are no acceptable guidelines for human exposure to aluminium in adults never mind in newborn infants and we have discussed many times the inadequacies of such published recommendations [6]. In the meantime, research continues to highlight the need to reduce exposure to aluminium in infants [7]. We do not know the form of aluminium in infant formulas and we can only speculate upon how much of this aluminium is absorbed across the infant gastrointestinal tract [6]. Until such much-needed research is available, precautions should be taken to reduce infant exposure to aluminium through formula feeding. All infant formula products reported upon herein were, as appropriate, reconstituted using ultrapure water. Formulas prepared in the home or elsewhere may use potable, as opposed to ultrapure, water in which the content of aluminium may additionally be high. Where possible, breast milk feeding should be prioritised, as the aluminium content of breast milk is invariably an order of magnitude lower than in formula feeds [7]. Where infant formulas are the only source of nutrition for many infants in their first weeks and months of life [11], aluminium ingested in formula feeds will be the major contributor to their body burden of aluminium. The last thing that vulnerable infants fed specialised formulas for their specific nutritional/medicinal need is additional aluminium in their diet. The encouraging news is that some of these prescription infant formulas are much less contaminated than their off-the-shelf counterparts and this highlights what can be achieved in reducing aluminium contamination of formula feeds. While prescription formulas are invariably more expensive than off-the-shelf products, this should not preclude future attempts to reduce their contamination and the contamination of infants by aluminium.

## Figures and Tables

**Table 1 ijerph-16-00899-t001:** Aluminium in ready-to-drink infant formulas designed for preterm and intrauterine growth restriction (IUGR) infants. Mean and SD are given, *n* = 5.

Brand	[Al] μg/L Mean (SD)	Al μg/Serving (Serving Size mL)
Cow & Gate Nutriprem 1	49.9 (33.7)	3.5 (70 mL)
Cow & Gate Nutriprem 2	139.3 (143.6)	27.9 (200 mL)
Cow & Gate Nutriprem Hydrolysed	167.1 (10.6)	15.0 (90 mL)
Danone Nutricia Infatrini Peptisorb	228.5 (48.3)	45.7 (200 mL)
SMA Pro First Infant Milk	249.4 (64.0)	17.5 (70 mL)

**Table 2 ijerph-16-00899-t002:** Aluminium in ready-to-drink infant formulas designed as supplements for infants struggling to gain weight. Mean and SD are given, *n* = 5.

Brand	[Al] μg/L Mean (SD)	Al μg/Serving (Serving Size mL)
Danone Nutricia Fortini Smoothie	709.6 (180.3)	141.9 (200 mL)
Danone Nutricia Fortini Multi Fibre	703.4 (53.7)	140.7 (200 mL)
Danone Nutricia Fortini Compact Multi Fibre Strawberry	568.2 (65.4)	71.0 (125 mL)
Danone Nutricia Fortini Compact Multi Fibre Neutral	784.5 (121.7)	98.1 (125 mL)
Nutrinovo ProSource TF Unflavoured	569.2 (18.1)	25.6 (45 mL)
Abbott Nutrition PediaSure Plus Juice Strawberry	153.5 (161.3)	30.7 (200 mL)
Abbott Nutrition PediaSure Plus Juice Apple	1956.3 (111.0)	391.3 (200 mL)
Nestlé Health Sciences Resource Fruit	180.2 (62.5)	36.0 (200 mL)

**Table 3 ijerph-16-00899-t003:** Aluminium in powdered formulas designed for infants with allergies and intolerances. Mean and SD are given, *n* = 5.

Brand	[Al] μg/g Mean (SD)	Al μg/Serving * Birth/6 Months	Al μg/Day * Birth/6 Months
SMA Nutrition Althera	0.46 (0.14)	6/14	53/69
Abbott Nutrition Similac Alimentum	1.65 (0.76)	12/38	na/na
Cow & Gate Pepti Junior	0.53 (0.40)	6/15	35/59
Nestlé Health Sciences Peptamen Junior	1.48 (0.24)	71 (no age spec)	na/na
Nutramigen Pregestimil Lipil	3.27 (0.19)	39/92	231/367
Danone Aptamil Pepti 1	0.35 (0.03)	4/12	26/47
SMA Nutrition Lactose Free	1.07 (0.15)	13/35	77/106

* Based upon manufacturer’s instructions.

**Table 4 ijerph-16-00899-t004:** Aluminium in powdered formulas supplemented with additional amino acids. Mean and SD are given, *n* = 5.

Brand	[Al] μg/g Mean (SD)	Al μg/Serving * Birth/6 Months	Al μg/Day * Birth/6 Months
SMA Nutrition Alfamino	0.27 (0.04)	4/8	21/24
Danone Nutricia Neocate LCP	0.29 (0.12)	4/9	24/47
Danone Nutricia Neocate Junior	0.61 (0.11)	19 (no age spec)	na/na
Nutramigen Puramino	2.23 (1.23)	28/64	167/256

* Based upon manufacturer’s instructions.

**Table 5 ijerph-16-00899-t005:** The concentration of aluminium in prescription formulas prepared as per the manufacturer’s instructions. Powdered formulas are identified in the table as bold script. Mean and SD are given, *n* = 5.

Brand	[Al] μg/L Mean (SD)
**SMA Nutrition** **Alfamino**	41.4 (6.1)
**Danone Nutricia** **Neocate LCP**	44.4 (18.4)
Cow & Gate Nutriprem 1	49.9 (33.7)
**Danone** **Aptamil Pepti 1**	52.5 (4.5)
**SMA Nutrition** **Althera**	67.5 (20.5)
**Cow & Gate** **Pepti Junior**	75.9 (57.3)
**Danone Nutricia** **Neocate Junior**	130.1 (23.6)
Cow & Gate Nutriprem 2	139.3 (143.6)
**SMA Nutrition** **Lactose Free**	**153.2 (21.5)**
Abbott Nutrition PediaSure Plus Juice Strawberry	153.5 (161.3)
Cow & Gate Nutriprem Hydrolysed	167.1 (10.6)
Nestlé Health Sciences Resource Fruit	180.2 (62.5)
Danone Nutricia Infatrini Peptisorb	228.5 (48.3)
**Abbott Nutrition** **Similac Alimentum**	**230.8 (106.3)**
SMA Nutrition Pro First Infant Milk	249.4 (64.0)
**Nestlé Health Sciences** **Peptamen Junior**	**325.6 (52.8)**
**Nutramigen** **Puramino**	**334.2 (184.3)**
**Nutramigen** **Pregestimil Lipil**	**468.2 (27.2)**
Danone Nutricia Fortini Compact Multi Fibre Strawberry	568.2 (65.4)
Nutrinovo ProSource TF Unflavoured	569.2 (18.1)
Danone Nutricia Fortini Multi Fibre	703.4 (53.7)
Danone Nutricia Fortini Smoothie	709.6 (180.3)
Danone Nutricia Fortini Compact Multi Fibre Neutral	784.5 (121.7)
Abbott Nutrition PediaSure Plus Juice Apple	1956.3 (111.0)

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
