# Peer review of "Prescription Infant Formulas Are Contaminated with Aluminium"

_ijerph, 2019, doi:10.3390/ijerph16050899_

Round 1

Reviewer 1 Report

IJERPH-D-19-451321:

Title: “Prescription Infant Formulas are Contaminated with Aluminium” by Redgrove et al.

General comments:

The paper addresses an interesting topic central to Al exposure in early life in the context of nursing infants that for medical reasons are deprived of the benefits of breastfeeding. Compared to other non-essential toxic elements, Al research receives far less attention. Additionally, the experience of this research in this line of research is notably wide and ground breaking. It is an interesting material worth of publication for many reasons.  

The material is well written, updates and adds new information to previous papers (ref 1 and 2) of this laboratory. My comments are made as additional information and should not be interpreted as any limitation of the study. Thus the authors should consider them as additional information to help understand Al exposure and toxicology in the context of this excellent material.

Pediatricians and clinical nutritionists are concerned about Al content in infant formulas; thus it is important to contextualize total Al load with other forms of exposure common in infancy such as Al exposure from vaccines (???), soy-based formula (???), and breast milk (???). Actually, breast-milk Al is only mentioned in passing (line 140). Although the studied formulas may be of interest to Europe, there are recent papers that have determined Al in human milk that should illustrate its variability.

Additionally, this also could be an opportunity to discuss bioavailability a fundamental part of interpreting levels of exposure and body loads assessment. Another point of interest/concern is feeding modes (breastfeeding and formula feeding). In breastfeeding, it has been reported (Chao et al, Pediatr Neonatol 2014;55:127-134) that there is a substantial decrease in Al concentrations from colostrum mature milk (56.5 μg/L to 13.4 μg/L). This is a physiological issue that need context; contrary to breastfeeding, formula feedings are not changeable with infants’ age and is usually taken in higher quantities than breast milk.

Specific comments:

Line 15:  “may be imported via ingredients, packaging and processing” how about in water during preparation of the powdered formulas?

Line 112:  … powdered products are generally less contaminated than ready-to-drink products (Table 5). I think somewhere near here the authors should draw attention that the final Al concentration will depend on the water available to prepare the powdered formulas.

Line 131:  “Table 5. The concentration of aluminium in prescription formulas prepared as per the manufacturer’s instructions.” What is the concentration of the water used?

Author Response

Reply to Reviewers

Comments from reviewers are shown in italics.

Reviewer 1

General comments:

The paper addresses an interesting topic central to Al exposure in early life in the context of nursing infants that for medical reasons are deprived of the benefits of breastfeeding. Compared to other non-essential toxic elements, Al research receives far less attention. Additionally, the experience of this research in this line of research is notably wide and ground breaking. It is an interesting material worth of publication for many reasons. 

Thank you

The material is well written, updates and adds new information to previous papers (ref 1 and 2) of this laboratory. My comments are made as additional information and should not be interpreted as any limitation of the study. Thus the authors should consider them as additional information to help understand Al exposure and toxicology in the context of this excellent material.

Thank you

Pediatricians and clinical nutritionists are concerned about Al content in infant formulas; thus it is important to contextualize total Al load with other forms of exposure common in infancy such as Al exposure from vaccines (???), soy-based formula (???), and breast milk (???). Actually, breast-milk Al is only mentioned in passing (line 140). Although the studied formulas may be of interest to Europe, there are recent papers that have determined Al in human milk that should illustrate its variability.

Infants in the age group of our study are exposed to aluminium in vaccines and other medications. However, we do not see any value in attempting to make comparisons between these different forms and routes of exposure and that of infant formulas and breast milk. It would simply be guesswork and speculation and certainly, where vaccines are concerned it would not make any sense to compare total burdens of aluminium by two disparate routes of exposure.  

Additionally, this also could be an opportunity to discuss bioavailability a fundamental part of interpreting levels of exposure and body loads assessment. Another point of interest/concern is feeding modes (breastfeeding and formula feeding). In breastfeeding, it has been reported (Chao et al, Pediatr Neonatol 2014;55:127-134) that there is a substantial decrease in Al concentrations from colostrum mature milk (56.5 μg/L to 13.4 μg/L). This is a physiological issue that need context; contrary to breastfeeding, formula feedings are not changeable with infants’ age and is usually taken in higher quantities than breast milk.

Again, we have no understanding at present of the biological availability of aluminium in infant formulas, or indeed breast milk. We do not consider that simple speculation of this matter is helpful. However, we can reveal that we are in the process of completing a clinical trial on this subject and we hope to be able to report on this very soon. We hope that the reviewer can bear with us on this.

Specific comments:

Line 15:  “may be imported via ingredients, packaging and processing” how about in water during preparation of the powdered formulas?

We use ultrapure water to prepare the infant formulas.

Line 112:  … powdered products are generally less contaminated than ready-to-drink products (Table 5). I think somewhere near here the authors should draw attention that the final Al concentration will depend on the water available to prepare the powdered formulas.

While the assertion by the reviewer is true one would hope that most have access to potable water that is not heavily contaminated with aluminium. However, we have added this proviso to the text.

Line 131:  “Table 5. The concentration of aluminium in prescription formulas prepared as per the manufacturer’s instructions.” What is the concentration of the water used?

See answer above.

Reviewer 2 Report

The topic of elemental contents of infant formula deserves attention. The manuscript can be improved by inclusion of a broader perspective and additional information/references.

The units do not appear correctly. For the purpose of review I have assumed that the units are expressed in micrograms (per liter, per gram, or per serving).

Abstract:

The level of Al in formula is reported as ranging from 49.9 (33.7) to 1956.3 (111.0) µg/L and 0.27 (0.04) to 3.27 (0.19) µg/g. It is not clear that the value in parenthesis represent SD. The values in parenthesis should be explained via a footnote or removed from the abstract.

Introduction:

Better context needs to be provided regarding the toxicity of aluminium to humans. Aluminium is one of, if not the most common element on earth and as such complete avoidance of exposure is impossible. Toxicity is a function of the level of exposure, as well as sensitivity of the individual(s) exposed. There are several sources that recommend ‘acceptable’ levels of human exposure. Among these sources are: US EPA PPRTV documentation (2006 https://hhpprtv.ornl.gov/issue_papers/Aluminum.pdf), which derives a chronic reference dose of 0.1 mg/kg-d. ATSDR (2008 https://www.atsdr.cdc.gov/toxprofiles/tp.asp?id=191&tid=34), which derived an oral intermediate and chronic minimal risk level of 1 mg/kg-d. Vanderplas et al 2014 also identifies 1 mg/kg-d as an acceptable tolerable level, citing FAO/WHO 2006.

While it is commonly accepted that elevated aluminium levels in parenteral and specialty formulas for pre-term infants or infants with compromising medical conditions are of concern there is not a consensus that current aluminium levels in infant formula represent a safety concern for healthy full term infants (e.g., Vanderplas et al 2014 Systematic review with meta-analysis).

Ljung Björklund et al 2012 have also reported that aluminium levels in breast milk of first time healthy mothers is higher than previously thought – but levels are also highly variable, with a mean (SD) value of 186 (584) µg/L and median of 86 µg/L (range was 21 to 4393 µg/L). This information should be provided in the Introduction as background information.

Aluminium is also used in water treatment. Powdered formulas are often reconstituted with tap water. The possible contribution of aluminium from water, in addition to what is in powdered formula itself, should also be mentioned.

Results:

This section is well written and presented. Table 3 and 4’s column title Al µg/feed needs additional context. The narrative identifies the data in this column as the amount per serving, which is the same title used in Tables 1 and 2. The data in Table 3 and 4 should be presented as µg/serving with the number of mls per serving in parenthesis as it is in Tables 1 and 2. In addition, a footnote should be added to explain how many servings per day were assumed to calculate µg/day from µg/feed.

Discussion:

It would be very useful to add a column showing the µg/kg per day ingested using the EFSA 2017 (http://www.efsa.europa.eu/en/efsajournal/pub/4849) recommended intake rate for assessing infant exposure. The resulting daily exposure can then be compared/contrasted to the reference values mentioned above as well as those resulting from breastmilk.

The Discussion section should also include mention of how formula type (e.g. cow’s milk, soy-based) can impact bioavailablity. An example, for manganese, of how elemental bioavailability can vary across formula types can be found in Valcke et al 2018 (https://www.mdpi.com/1660-4601/15/6/1293).

Conclusion:

The conclusion should include mention of available reference values regarding the limited number and type of tolerable intake rates (e.g., PPRTV, ATSDR) as well as the recent levels in breastmilk reported by Ljung Björklund et al 2012.

Author Response

Reply to Reviewers

Comments from reviewers are shown in italics.

Reviewer 2

The topic of elemental contents of infant formula deserves attention. The manuscript can be improved by inclusion of a broader perspective and additional information/references.

The units do not appear correctly. For the purpose of review I have assumed that the units are expressed in micrograms (per liter, per gram, or per serving).

Apologies, this is due to the publisher and not us!

Abstract:

The level of Al in formula is reported as ranging from 49.9 (33.7) to 1956.3 (111.0) µg/L and 0.27 (0.04) to 3.27 (0.19) µg/g. It is not clear that the value in parenthesis represent SD. The values in parenthesis should be explained via a footnote or removed from the abstract.

Once you have the original version of the manuscript with all units correct then we do not think that there is any need to explain that data have been given as mean and SD each time.

Introduction:

Better context needs to be provided regarding the toxicity of aluminium to humans. Aluminium is one of, if not the most common element on earth and as such complete avoidance of exposure is impossible. Toxicity is a function of the level of exposure, as well as sensitivity of the individual(s) exposed. There are several sources that recommend ‘acceptable’ levels of human exposure. Among these sources are: US EPA PPRTV documentation (2006 https://hhpprtv.ornl.gov/issue_papers/Aluminum.pdf), which derives a chronic reference dose of 0.1 mg/kg-d. ATSDR (2008 https://www.atsdr.cdc.gov/toxprofiles/tp.asp?id=191&tid=34), which derived an oral intermediate and chronic minimal risk level of 1 mg/kg-d. Vanderplas et al 2014 also identifies 1 mg/kg-d as an acceptable tolerable level, citing FAO/WHO 2006. While it is commonly accepted that elevated aluminium levels in parenteral and specialty formulas for pre-term infants or infants with compromising medical conditions are of concern there is not a consensus that current aluminium levels in infant formula represent a safety concern for healthy full term infants (e.g., Vanderplas et al 2014 Systematic review with meta-analysis).

The reviewer may not be aware that one of us (CE) has spent the last 35 years trying to understand human exposure to aluminium and its possible toxicity. CE has written about so-called ‘safe- limts’ proposed by ‘committee’ and thorough investigation of the bases for these ‘limits’ has revealed that they are ‘straw’ data built on almost no relevant studies. They are meaningless values obtained simply because ‘policy’ demands them.

We apologise to the reviewer but CE having spent a lifetime in science criticising these data in print we are not now going to give them credence by including them in this study.

Ljung Björklund et al 2012 have also reported that aluminium levels in breast milk of first time healthy mothers is higher than previously thought – but levels are also highly variable, with a mean (SD) value of 186 (584) µg/L and median of 86 µg/L (range was 21 to 4393 µg/L). This information should be provided in the Introduction as background information.

This study is not about breast milk nor is about comparing aluminium in infant formula with aluminium in breast milk. However, we have used Web of Science to try to find the paper by Ljung Björklund et al. 2012 and it is not in this database.

Aluminium is also used in water treatment. Powdered formulas are often reconstituted with tap water. The possible contribution of aluminium from water, in addition to what is in powdered formula itself, should also be mentioned.

Thank you, we have included reference to this in the revised manuscript.

Results:

This section is well written and presented. Table 3 and 4’s column title Al µg/feed needs additional context. The narrative identifies the data in this column as the amount per serving, which is the same title used in Tables 1 and 2. The data in Table 3 and 4 should be presented as µg/serving with the number of mls per serving in parenthesis as it is in Tables 1 and 2. In addition, a footnote should be added to explain how many servings per day were assumed to calculate µg/day from µg/feed.

Thank you, we have changed this as suggested.

 Discussion:

It would be very useful to add a column showing the µg/kg per day ingested using the EFSA 2017 (http://www.efsa.europa.eu/en/efsajournal/pub/4849) recommended intake rate for assessing infant exposure. The resulting daily exposure can then be compared/contrasted to the reference values mentioned above as well as those resulting from breastmilk.

Apologies but as stated previously we prefer not to do this. The EFSA limit is scientifically meaningless.

The Discussion section should also include mention of how formula type (e.g. cow’s milk, soy-based) can impact bioavailablity. An example, for manganese, of how elemental bioavailability can vary across formula types can be found in Valcke et al 2018 (https://www.mdpi.com/1660-4601/15/6/1293).

May we refer the reviewer to our answer to Reviewer 1 on this subject. We see no reason to simply speculate upon this until there are useful data relating to aluminium.

Conclusion:

The conclusion should include mention of available reference values regarding the limited number and type of tolerable intake rates (e.g., PPRTV, ATSDR) as well as the recent levels in breastmilk reported by Ljung Björklund et al 2012.

Thank you but as we have already explained we do not agree with these non-scientific policy limits.